# Co-Culture Strategy of *Lactobacillus kefiranofaciens* HL1 for Developing Functional Fermented Milk

**DOI:** 10.3390/foods10092098

**Published:** 2021-09-05

**Authors:** Sheng-Yao Wang, Ren-Feng Huang, Ker-Sin Ng, Yen-Po Chen, Jia-Shian Shiu, Ming-Ju Chen

**Affiliations:** 1Department of Animal Science and Technology, National Taiwan University, Taipei 10617, Taiwan; yaoyao@ntu.edu.tw (S.-Y.W.); xd2592008@gmail.com (R.-F.H.); r04626001@g.ntu.edu.tw (K.-S.N.); 2Department of Animal Science, National Chung Hsing University, Taichung 40227, Taiwan; chenyp@dragon.nchu.edu.tw; 3The iEGG and Animal Biotechnology Center, National Chung Hsing University, Taichung 40227, Taiwan; 4Hengchun Branch, Livestock Research Institute, Council of Agriculture, Executive Yuan, Pingtung 94644, Taiwan; Wadeshiu@mail.tlri.gov.tw

**Keywords:** *Lactobacillus kefiranofaciens*, probiotics, co-culture, fermented skimmed milk

## Abstract

Our previous studies indicated that *Lactobacillus kefiranofaciens* HL1, isolated from kefir grain, has strong antioxidant activities and anti-aging effects. However, this strain is difficult to use in isolation when manufacturing fermented products due to poor viability in milk. Thus, the purpose of this study was to apply a co-culture strategy to develop a novel probiotic fermented milk rich in *L. kefiranofaciens* HL1. Each of four selected starter cultures was co-cultured with kefir strain HL1 in different media to evaluate their effects on microbial activity and availability of milk fermentation. The results of a colony size test on de Man, Rogosa and Sharpe (MRS) agar agar, microbial viability, and acidification performance in MRS broth and skimmed milk suggested that *Lactococcus lactis* subsp. *cremoris* APL15 is a suitable candidate for co-culturing with HL1. We then co-cultured HL1 and APL15 in skimmed milk and report remarkable improvement in fermentation ability and no negative impact on the viability of strain HL1 or textural and rheological properties of the milk. Through a co-culture strategy, we have improved the viability of kefir strain HL1 in fermented skimmed milk products and successfully developed a novel milk product with a unique flavor and sufficient probiotics.

## 1. Introduction

Fermented dairy products have been recognized as healthy foods for thousands of years. It is well-known that the fermentation process can extend the shelf-life of fresh milk and nourish the flavor of the final product with high nutrition value. For industrial commercialization, two starter cultures, *Lactobacillus delbrueckii* subsp. *bulgaricus* (*L. delbrueckii* subsp. *bulgaricus*, LB) and *Streptococcus thermophilus* (*S. thermophilus*, ST), are commonly used to produce fermented milk [1], defined as “yogurt” [2]. Other lactic acid bacteria (LAB), such as *Lactobacillus* (*L. acidophilus*, *L. johnsonii*, *L. reuteri*, and *L. rhamnosus*) and *Bifidobacterium* (*Bifidobacterium bifidum*, *B. breve*, *B. infantis*, and *B. longum*), are also introduced to increase the variety and health benefits of dairy products [3,4].

However, the use of LAB as probiotics in the development of high-quality fermented milk is a challenging task. One of the crucial requirements is to maintain a sufficient number of probiotic cells throughout the manufacturing process and shelf life with no adverse effects on the flavor, aroma, and post-acidification of the final products [5]. Many studies have indicated that the survival of *L. acidophilus* and genus *Bifidobacterium* were decreased in fermented milk due to the accumulation of organic acids and hydrogen peroxide [6,7]. Besides, the fermentation characteristics of probiotics in milk, such as acidification time, appropriate taste and aromatic profiles, and tolerance to food additives, should also be considered [3,8,9].

Nevertheless, co-culture strategies might provide solutions to tackle the challenges for the utilization of probiotic strains in fermented dairy products. The communications between *Lactobacillus* and *Saccharomyces cerevisiaei* through metabolites, microbial aggregation, and biofilm formation could increase microbial counts and organic acids in final products, prevent the contamination from spoilage bacteria during fermentation, and enhance exopolysaccharide production [10]. Recently, Xu et al. [11] thoroughly reviewed the coexistence-relevant mechanisms and molecular regulatory network when co-culturing *Lactobacillus* with *S*. *cerevisiae* in various fermented products. In addition to co-culture strategies between *lactobacillus* and yeast, fermentation of yogurt starter cultures with *L. plantarum* could enhance the consumption of sugar in milk for developing acceptable low-sugar yogurt [12]. Casarotti et al. [13] found that acidification rates of *B. animalis* subsp. *lactis* and *L. acidophilus* were improved by 32% and 74%, respectively, when co-cultured with *S. thermophilus* (ST) in reconstituted milk. The presence of ST also increased the viable cells of *L. acidophilus* after 28 days of storage [13] and inhibited the acetic acid production of *B. animalis* subsp. *lactis* to avoid the unpleasant vinegar-like flavor in the dairy product [14]. In another study, co-culturing ST with *B. lactis* could increase the biomass of the latter by about 38% and enhance the diacetyl compound in milk as compared with pure culture [15]. However, Ranadheera et al. [16] indicated that different co-culturing combinations of probiotics such as *L. acidophilus*, *B. animalis* subsp. *Lactis*, and *Propionibacterium jensenii* in goat milk was unable to provide positive effects on the sensory properties due to possible development of unpleasant flavor, organic acid, and unstable curd in fermented goatmilk products during fermentation and storage [17]. Therefore, it is critical to understand the influence of probiotics and starter cultures on the off-flavor of fermented milk by detecting volatile compounds [18] and then select suitable and desirable microorganisms for co-culture strategies. In terms of improving functional properties, ST co-cultured with *L. plantarum* or *B. animalis* ssp. *lactis* could intensify antioxidant capacity and ACE inhibition activity of the fermented milks [19]. Additionally, certain strains of ST as co-cultured with *L. brevis* could stimulate *L. brevis* to produce the neurotransmitter, γ-amino-butyric acid (GABA) [20]. Khanlari et al. [21] also demonstrated *Enterococcus faecium* had a greater acid-producing ability and significantly produced higher amounts of GABA when co-culturing with *Lc. lactis* subsp. *lactis* in milk. These findings suggest that a co-culture strategy is applicable to improve the viability, organoleptic characteristics, and functional properties of probiotic microorganisms in dairy products.

Previously, *L. kefiranofaciens* HL1, an exopolysaccharide producer [22], was isolated from a Taiwanese kefir grain in our lab. This strain has demonstrated in vitro antioxidant activity by measuring the inhibition of linoleic acid peroxidation, chelation ability for Fe^2+^, and 1, 1-diphenyl-2-picrylhydrazyl (DPPH) scavenging activity. The further in vivo study by D-galactose-induced aging mice demonstrated that daily administrating of *L. kefiranofaciens* HL1 found that the HL1 strain exhibits anti-aging properties by strengthening the resistance to oxidative stress, improving memory and learning abilities, and modulating the composition of gut microbiota [23,24]. However, *L. kefiranofaciens* HL1 has a low growth rate and poor viability during milk fermentation, which obstructed the usage of this unique strain. Co-culture strategy might provide a solution, but little information is available on the effects of various lactic acid bacteria on *L. kefiranofaciens* during fermentation. Thus, in the present study, we evaluated the synergistic effects of four selected mesophilic and thermophilic starter strains with *L. kefiranofaciens* HL1 in milk. The goal of this study was to develop a novel probiotic fermented milk with good fermentation parameters and rheological and sensory properties.

## 2. Materials and Methods

### 2.1. Bacteria Cultures

*Lactobacillus kefiranofaciens* HL1 (*L. kefiranofaciens* HL1) and *Lactococcus lactis* subsp. *cremoris* APL15 (*Lc. lactis* subsp. *cremoris* APL15) were previously isolated from Taiwanese kefir grain and Taiwanese ropy fermented milk, respectively [25,26,27]. *L. delbrueckii* subsp. *bulgaricus* BCRC 10696^T^, *Streptococcus thermophilus* BCRC 12268 (*S. thermophilus* BCRC 12268), and *S. thermophilus* BCRC 13889^T^ were purchased from Bioresource Collection and Research Center (BCRC) of the Food Industry Research and Development Institute (FIRDI, Hsinchu, Taiwan). All bacteria strains were grown in MRS broth (Acumedia Manufacture, Lansing, MI, USA) at 30 °C with 1% inoculation. The cultures were activated twice before further experiments.

### 2.2. Screening Candidate Strains for Co-Culture with L. kefiranofaciens HL1

#### 2.2.1. Colony Size on Agar Plate

The method was modified as described by Sieuwerts et al. [28]. *L. kefiranofaciens* HL1 was diluted and spread on MRS agar. The candidate strains were then inoculated at the four corners and in the center of the plate (2 μL/spot). *L. delbrueckii* subsp. *bulgaricus* BCRC 10696^T^, *S. thermophilus* BCRC 12268, *S. thermophilus* BCRC 13889^T^, and *Lc. lactis* subsp. *cremoris* APL15 were selected as candidate strains. Sterilized 0.85% saline solution was used as the control. After anaerobic cultivation at 30 °C for 72 h, the plate with the appropriate dilution of *L. kefiranofaciens* HL1 (20–40 colonies) was photographed, and the average size of colonies was calculated by Image J. The result was presented as relative colony size in percentage by using control as a baseline.

#### 2.2.2. Cultivation with Supernatant of Candidate Strain

For the candidate strain that decreased the relative colony size of *L. kefiranofaciens* HL1, the effect of its supernatant on the growth of *L. kefiranofaciens* HL1 was evaluated. First, activated culture was centrifuged at 1000× *g*, for 10 min, and the supernatant was collected and filtered (Millex-GV Filter, 0.22 μm, Merck KGaA, Darmstadt, Germany). Ten percent of filtered supernatant was added to MRS broth and inoculated with 1% of *L. kefiranofaciens* HL1. After cultivation at 30 °C for 24 h, 1 mL of culture broth was serially diluted with 0.85% saline solution and plated on MRS agar. The colonies were counted and expressed as colony-forming per unit (CFU/mL) after incubation at 30 °C for 24 h.

### 2.3. Co-Culture Conditions in MRS Broth and Skimmed Milk

Each candidate strain (1% inoculation, ~10^6^ CFU/mL) was co-cultured with *L. kefiranofaciens* HL1 (1% inoculation, ~10^6^ CFU/mL) in MRS broth and 10% (g/mL) skimmed milk, separately, at 30 °C for 24 h. Microbial count of *L. kefiranofaciens* HL1 was counted with acidified MRS agar (pH 5.2, adjusted with 1N hydrochloric acid) after anaerobic incubation at 30 °C for 72 h, while the other strains were calculated with M17 agar (Acumedia Manufacture, Lansing, MI, USA) after aerobic incubation at 30 °C for 24 h. Besides, the pH value of cultivated broth was also evaluated with a Lab 850 pH meter (SI Analytics GmbH, Berlin, Germany).

### 2.4. Production of Fermented Milk

*L. kefiranofaciens* HL1 and selected candidate strains were inoculated in 10% (g/mL) skimmed milk (each at 1%, ~10^6^ CFU/mL), and incubated at 30°C for preparing fermented milk samples. Commercial yogurt was made by skimmed milk fermented with commercial starter culture YC-380 (a combination of *L. delbrueckii* subsp. *bulgaricus* and *S. thermophilus*; Chr. Hansen Holding A/S, Hoersholm, Denmark). GDL-induced curd was produced using 1.5% glucono delta-lactone (GDL). When the pH values of all experimental groups were below 4.50 ± 0.05, fermentation and acidification processes were stopped. The microbial assessment was carried out as previously described. After fermentation and acidification, the samples were stored at 4 °C for 24 h for further syneresis, textural, and rheological analyses.

#### 2.4.1. Physicochemical Properties

The acidity of fermented milk was determined based on ISO 6901:2010, using 0.5 mL of 1% (g/mL in absolute alcohol) phenolphthalein as an indicator for titration. Syneresis was performed according to Mani-López et al. [29]. The textural analysis was determined by TA.XT plus Texture Analyser (Stable Micro Systems, Surrey, UK) with a 5 kg load cell and an A/BE back extrusion cell at the following settings: test speed, 1 mm/s; post-test speed, 1 mm/s; distance, 25 mm; and the rate for data acquisition, 200 points per s (pps) [30]. The firmness, consistency, cohesiveness, and viscosity index of fermented milk samples were calculated by positive peak force, area of positive region, peak negative force, and area of negative region, respectively. For viscosity, the tested samples were gently stirred 20 times and set still for 5 min to allow rebuilding gel structure [31]. RST-CPS Touch Rheometer (Brookfield Engineering Laboratories Inc., MA, USA) with a spindle of RPT-50 (using parallel geometry at 1 mm gap) was used to record the apparent viscosity of the fermented milk samples for 300 s under constant rotation speed of 30 rpm at 4 °C [32].

#### 2.4.2. Sensory Evaluation

Sensory evaluation was conducted by performing a hedonic scale test on 30 mL of fermented milk products (4 °C) from each treated sample. The evaluation was carried out by 30 semi-trained panelists comprised of students and faculty members in the Department of Animal Science and Technology at the National Taiwan University, who were familiar with dairy products. A total of 17 men (56.67%) and 13 women (43.33%) with age between 20 to 31 years took part in this event. Three kinds of fermented milk samples were stored at 4 °C for 24 h and were placed in 50 mL plastic cups coded individually with random three-digit numbers. The samples were tested in a random order and all evaluations were performed at room temperature. Nine-level hedonic tests (1, dislike extremely; 2, dislike very much; 3, dislike; 4, dislike slightly; 5, neither dislike nor like; 6, like moderately; 7, like; 8, like very much; 9, like extremely) in terms of appearance, aroma, texture, flavor, and overall evaluation were completed by the panelists [33].

### 2.5. Statistical Analysis

Sensory evaluation was accessed with non-parametric statistics methods, including the Kruskal–Wallis test and Dunn’s test, while the other experiments (three replicates) were analyzed using the ANOVA GLM procedure in Statistical Analysis Systems (SAS) software. Comparisons between two groups and multiple groups were processed with student’s *t*-test and Tukey’s HSD (honest significant difference) test, respectively.

## 3. Results and Discussion

### 3.1. Pre-Screening Suitable Starter Cultures for Co-Culturing with Probiotic HL1

To select a suitable bacteria strain for co-culturing with *L. kefiranofaciens* HL1 to produce probiotic fermented milk, we first evaluated the effects of four LAB strains often applied as starter cultures in dairy products on the growth of *L. kefiranofaciens* HL1 incubated in MRS medium. We found that (Figure 1a) the total colony areas of L. *kefiranofaciens* HL1 solely grown on MRS agar plate (control group) were not significantly different from those co-cultured with *S. thermophilus* BCRC 12268 (ST 12268), *S. thermophilus* BCRC 13869^T^ (ST 13869), and *Lc. lactis* subsp. *cremoris* APL15. These three strains also showed no influence on the colony-forming rate of *L. kefiranofaciens* HL1 during incubation, indicating their practicability for co-culturing with *L. kefiranofaciens* HL1. In contrast, the growth of *L. kefiranofaciens* HL1 was suppressed by the presence of *L. delbrueckii* subsp. *bulgaricus* BCRC 10696^T^ (LB 10696) with a significant reduction in the relative colony size of *L. kefiranofaciens* HL1 on the MRS plate compared with the control group (*p* < 0.05). To verify the inhibitory phenomenon, *L. kefiranofaciens* HL1 was inoculated in MRS broth with the additional 10% filtered supernatant from cultures of LB 10696 (Figure 1b). The viable bacterial counts of *L. kefiranofaciens* HL1 in MRS broth with 10% filtered supernatant from cultures of LB 10696 were significantly lower during 12 or 24 h of incubation as compared with the non-supernatant counterpart (*p* < 0.05). Whereas no adverse effect on the cell counts with additional 10% filtered supernatants from the other three cultures was observed (data not shown). Although LB 10696 was isolated from Bulgarian yogurt and is widely used in dairy fermentation to produce commercial yogurt and cheese, their metabolites in supernatant might cause an adverse effect on the viabilities and colony size of *L. kefiranofaciens* HL1. Our findings were paralleled with previous studies reporting that the activity and viable counts of probiotics belonging to genus *Lactobacillus* in mix-culture fermented products are decreased by certain *L. delbrueckii* subsp. *bulgaricus* strains through nutritional competition and the mutual inhibition of metabolites such as peroxides and organic acid [7,20,29]. Since LB 10696 inhibited the growth of *L. kefiranofaciens* HL1, this strain was deleted from candidate starters in this study.

Another three candidate starter cultures, including ST 12268, ST 13869, and *Lc. lactis* subsp. *cremoris* APL15 (APL15), were further studied in symbiosis with L. *kefiranofaciens* HL1 in MRS medium. For the viabilities of *L. kefiranofaciens* HL1 (Figure 2a), no significant change among groups was observed as co-culturing with ST 12268, ST 13869, and APL15 in the MRS at 30 °C for 12 and 24 h. The viabilities of other starter strains also showed no statistical difference when co-culturing with *L. kefiranofaciens* HL1 for 24 h (data not shown).

The acid-producing ability of starter cultures is also important when producing fermented milk products. During 12- and 24-h incubation, the pH values of the HL1 group in the MRS were 5.86 ± 0.07 and 5.14 ± 0.07, respectively (Figure 3a). All co-culturing groups had significantly lower pH compared with the HL1 group (*p* < 0.05), except the HL1+ST 12268 group for 12-h incubation. It is worth noting that the HL1+APL15 group showed a significantly lower pH than the HL1 and APL15 counterparts in MRS broth, indicating that the co-culture strategy for *Lc. lactis* subsp. *cremoris* APL15 and *L. kefiranofaciens* HL1 could provide a positive effect to stimulate each other to produce organic acids. These findings indicated that *Lc. lactis* subsp. *cremoris* APL15 might have stimulated probiotic *L. kefiranofaciens* HL1 to produce organic acids during co-incubation in MRS medium.

### 3.2. Evaluation of Co-Culture in Skimmed Milk

After an investigation using MRS broth, we used skimmed milk to select the co-culturing starter cultures. For viability, *L. kefiranofaciens* HL1 could increase approximately 1.5 log CFU/mL after 24-h incubation in skimmed milk (Figure 2b). There were no significant differences in viable bacterial counts among the HL1, HL1+ST 13869, and HL1+APL15 groups. In contrast, the viable bacterial counts of *L. kefiranofaciens* HL1 were significantly suppressed from 6.0 log CFU/mL to 5.5 log CFU/mL when co-culturing with ST 12268 in skimmed milk during fermentation (*p* < 0.05). Interestingly, this inhibitory phenomenon was not observed in the MRS medium. Another strain, ST 13869, also showed no negative impact on the viability of *L. kefiranofaciens* HL1 during co-culture in skimmed milk. Previous studies have demonstrated that different *S. thermophilus* strains show diverse effects on the viability of LAB [12,15]. Certain *S. thermophilus* strains could produce specific nutrients, such as formic acid, folic acid, fatty acids, and amino acids during milk fermentation for promoting the growth of the *Lactobacillus* genus [34,35,36]. Whereas Fontaine and Hols [37] reported that *S. thermophilus* LMD-9 produced bacteriocin-like peptides against Gram-positive bacteria and inhibited the growth of nonstarter strains or food-borne pathogen bacteria. The different growth mediums could also influence the production and stability of bacteriocin-like peptides released by some *S. thermophilus* strains [38]. This finding suggested that the importance for successful application of co-culture strategy is not only dependent on microbial strains; the fermented medium is also a crucial factor. Further studies are required to clarify the inhibitory materials of ST 12268 and relative mechanisms.

Regarding the effect of *L. kefiranofaciens* HL1 on the bacterial counts of three starter cultures, only *Lc. lactis* subsp. *cremoris* APL15 had a significantly lower bacteria count (*p* < 0.05) when co-culturing with *L. kefiranofaciens* HL1 than that without co-culturing for both 12- and 24-h fermentation (Figure 4). Although both ST strains showed no effect when co-culturing with *L. kefiranofaciens* HL1, *L. kefiranofaciens* has been reported to possess an antimicrobial ability [39]. *L. kefiranofaciens* DD2 could inhibit certain causative bacteria due to suppression of biofilm formation-associated genes, which are related to carbohydrate metabolism, biofilm formation, and adhesion proteins [40]. In fact, *Lc. lactis* subsp. *cremoris* APL15 is an exopolysaccharide producer used to increase the ropy and adhesion characteristics of fermented products. The secretion of exopolysaccharides was associated with carbohydrate metabolism and biofilm formation [41]. Thus, the presence of *L. kefiranofaciens* HL1 in the co-culture system might change the carbohydrate metabolites of *Lc. lactis* subsp. *cremoris* and further suppress the bacterial counts.

For acid-producing ability, HL1 co-culturing with ST 12268, ST 13869, and APL 15 could significantly decrease the pH values of the fermented milk samples for both 12- and 24-h incubations (Figure 3b) (*p* < 0.05). Since *L. kefiranofaciens* HL1 had a poor acid-producing ability in skimmed milk, the co-culture strategy could help this unique strain to grow and produce acid in fermented milk. In fact, the composition of inoculated bacterial strains highly impacts the acidification of the fermented milk. Sodini et al. [42] demonstrated that some probiotic bacteria grew weakly in milk. This phenomenon was accompanied by poor milk acidification. In contrast, the starter strains proliferated quickly and had a positive effect on acidification during milk fermentation. Thus, a co-culture strategy with starter cultures could provide a good opportunity for the commercial production of probiotic fermented milk. Among three starter strains in our study, *Lc. lactis* subsp. *cremoris* APL15 presented the highest acid-production. *Lc. lactis* subsp. *cremoris* was also a potential probiotic strain due to its exopolysaccharides-producing characteristic and health benefits [43,44,45]. The findings suggested that *Lc. lactis* subsp. *cremoris* APL15 was the best candidate starter strain for co-culturing with *L. kefiranofaciens* HL1 to develop multiple functional fermented milks. Therefore, we applied a co-culture strategy to develop a fermented milk product with HL1 and APL15 and determine its microbial, physicochemical, and sensory properties.

### 3.3. Physicochemical and Sensory Properties of Fermented Milk with HL1

#### 3.3.1. pH Value and Titratable Acidity

The results of pH profiles during milk fermentation at 30 °C (Figure 5a) showed that fermented skimmed milk prepared by culturing with *Lc. lactis* subsp. *cremoris* APL15 alone (FSM^APL15^) and co-culturing *L. kefiranofaciens* HL1 with *Lc. lactis* subsp. *cremoris* APL15 (FSM^HL1+APL15^) had a significantly lower pH after 12-h fermentation than the HL1 fermented skimmed milk (FSM^HL1^) (*p* < 0.05). After 24-h fermentation, FSM ^HL1+APL15^ showed significantly lower pH and higher titratable acidity than the other two groups (Figure 5a,b) (*p* < 0.05), and meets the titratable acidity requirement of the Food and Agriculture Organization of the United Nations (FAO) [2] for fermented milk products. Oliveira et al. [46] demonstrated that milk acidification by co-culture of a probiotic strain with a starter culture outperformed the probiotic alone. This might be because the co-culture strategy could improve microbial fermentation ability and increase the acid level of products to create a suitable probiotic fermented milk.

#### 3.3.2. Syneresis and Textural Analysis

For commercial purposes, it is important to maintain the stability and structure of acid-induced milk curd during shipping and storage. Thus, the syneresis and total textural profiles of the acid-induced milk curds by fermentation or acidification (GDL) were analyzed. Among the four samples, GDL-induced curd showed the highest syneresis (*p* < 0.05) (Table 1). Both FSM^APL15^ and FSM^HL1+APL15^ had lower syneresis compared with the GDL group. As for physical and textural profiles, FSM^HL1+APL15^ showed a trend to increase firmness, consistency, cohesiveness, and resistance to syneresis as compared with other acid-induced milk curds.

We further measured the rheological properties. The apparent viscosity of four curd samples demonstrated a similar pattern, which decreased with an increase in shear time at a constant temperature and rotation speed (Figure 5c). Both FSM^APL15^ and FSM^HL1+APL15^ had higher apparent viscosity than other groups, even after stirring, which was consistent with our previous results in physical and textural profiles. We noticed that fermented milk made by starter strain of *Lc. lactis* subsp. *cremoris* APL15 provided better physical and stable properties than non-fat yogurt or GDL-induced milk curd. Kristo et al. [47] demonstrated that applying the ropy strain of *Lc. lactis* subsp. *cremoris* JFR1, when producing fermented milk, could increase the storage modulus and viscosity due to the EPS production. Moreover, the secretion of EPS by *Lc. lactis* subsp. *cremoris* JFR1 could reduce the recovery of the protein–protein interaction networks after shearing, and the viscoelastic acid-induced gel would become stir form with ropy stable semisolids. In terms of the dairy industry and consumer market, syneresis or whey separation is an important defect in fermented milk [48]. Our finding suggests that starter strain APL15 might play a crucial role associated with the production of sufficient acid and EPS to form a firm and stable fermented milk product with less syneresis. The EPS produced by *Lc. lactis* subsp. *cremoris* APL15 was a great in situ natural stabilizer that prevents syneresis, strengthens the gel structure during processing and storage, and provides sensory properties.

#### 3.3.3. Sensory Evaluation

To understand how these bacteria affected consumer acceptability and sensory scores of fermented milk, three kinds of fermented milk, including non-fat yogurt, FSM^APL15^, and FSM^APL15+HL1^, were compared and scored in this study. Sensory scores for the appearance of all samples were approximately 6.29–6.82 with no significant difference (*p* > 0.05) (Table 2). The fermented milk produced by inoculation with each culture were uniform in appearance since all of them formed intact curds without appearance defects, such as whey separation and coarse surface. The texture scores were also similar and ranged from 5.28 to 5.86. The sensory analysis of appearance and texture were consistent with the results of texture profiles in Table 2, and there was no significant difference among the three kinds of fermented milk. However, the scores for aroma, flavor, and overall acceptability of FSM^HL1+APL15^ were significantly lower than that of other samples. *L. kefiranofaciens* HL1 participated in milk fermentation would decrease the organoleptic properties of final fermented products as compared with FSM^APL15^. Walsh et al. [49] analyzed the causal relationship between microbial taxa and volatile compounds in kefir fermentation and found that *L. kefiranofaciens* correlated with carboxylic acids and ketones associated with cheesy flavors and with esters associated with fruity flavors. Several researchers indicated that certain probiotics would cause unpleasant flavor in products via producing different levels and kinds of metabolic compounds that might decrease the consumer palatability [50,51,52]. Therefore, it will be necessary to improve the aroma and flavor of FSM^HL1+APL15^ to increase the total acceptance and consumption. Junaid et al. [53] indicated that probiotic fermented milk produced by adding different flavors, such as strawberry, pineapple, and mango, possessed higher overall acceptability. Moreover, the incorporation of natural fruits or fruit pulps into the probiotic fermented dairy products is an alternative means to improve the sensory profiles [54,55], and would cause our probiotic fermented milk to possess a combination of natural, delicious, and health-promoting properties.

## 4. Conclusions

In the present study, *Lc. lactis* subsp. *cremoris* APL15 is identified as a superior starter culture for enhancing acid production rate and keeping *L. kefiranofaciens* HL1 vitality of the fermented ecosystem either in MRS broth or skimmed milk. The co-fermentation of skimmed milk with *Lc. lactis* subsp. *cremoris* APL15 and *L. kefiranofaciens* HL1 for making fermented milk products is a better strategy to provide sufficient probiotic counts and better physicochemical properties with less whey separation. However, improvement of the aroma and flavor of this fermented milk is necessary to increase consumer acceptability. To the best of our knowledge, this is the first report developing a novel probiotic fermented milk using critical LAB isolated from Taiwanese kefir grains and Taiwanese ropy fermented milk.

## Figures and Tables

**Figure 1 foods-10-02098-f001:**
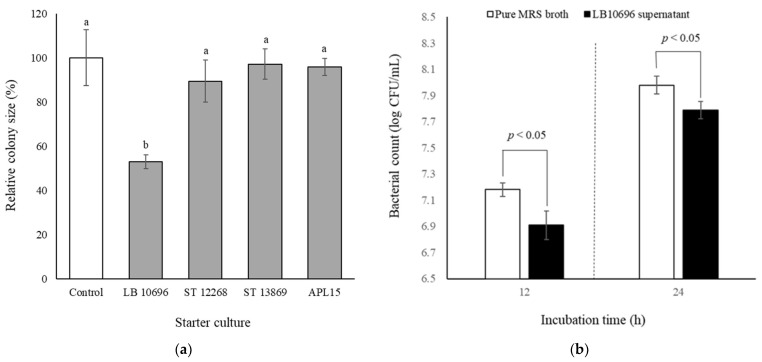
Effect of starter cultures on the growth of *L. kefiranofaciens* HL1 incubated in MRS medium. (**a**) Relative colony size of *L. kefiranofaciens* HL1 when culturing alone (control) and co-culturing with different starter cultures on MRS agar at 30 °C for 72 h. Data are given as mean ± SD (*n* = 3). ^a,b^ bars without the common letter are significantly different (*p* < 0.05). (**b**) Viable bacterial count of *L. kefiranofaciens* HL1 in pure MRS broth and in MRS broth with 10% filtered supernatant from cultures of LB 10696 at 30 °C for 12 and 24 h. Data are given as mean ± SD (*n* = 3). Abbreviations: LB 10696, *L. delbrueckii* subsp. *bulgaricus* BCRC 10696T; ST 12268, *S. thermophilus* BCRC 12268; ST 13869, *S. thermophilus* BCRC 13869T; APL15, *Lc. lactis* subsp. *cremoris* APL15.

**Figure 2 foods-10-02098-f002:**
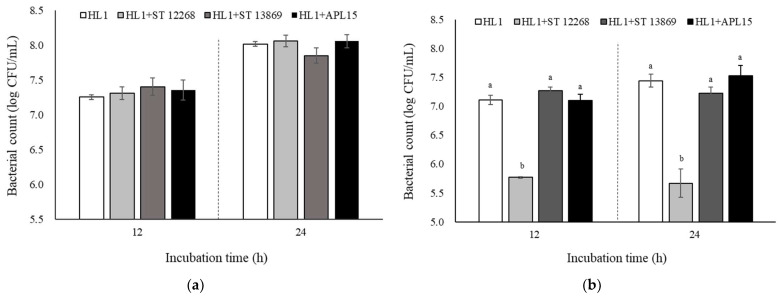
Effect of HL1 co-culturing with each selected strain on HL1 bacterial counts in MRS broth (**a**) and skimmed milk (**b**) at 30 °C for 12 and 24 h. Data are given as mean ± SD (*n* = 3). ^a,b^ Mean values without the common letter within the same incubation time indicate a significant difference (*p* < 0.05). Abbreviations: HL1, *L. kefiranofaciens* HL1; ST 12268, *S. thermophilus* BCRC 12268; ST 13869, *S. thermophilus* BCRC 13869^T^; APL15, *Lc. lactis* subsp. *cremoris* APL15.

**Figure 3 foods-10-02098-f003:**
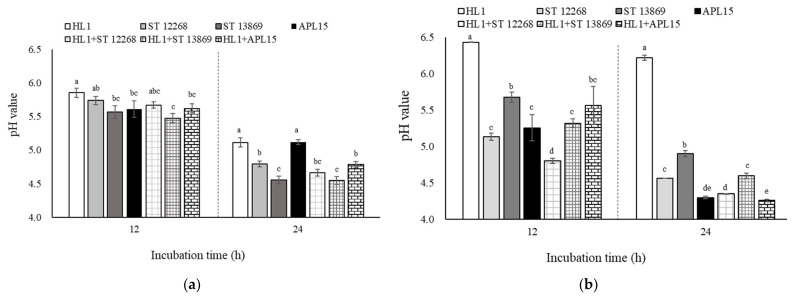
Effect of HL1 co-culturing with each selected strain on pH values of MRS broth (**a**) and skimmed milk (**b**) for both 12- and 24-h fermentation. Data are given as mean ± SD (*n* = 3). ^a–e^ Mean values without the common letter within the same incubation time indicate a significant difference (*p* < 0.05). Abbreviations: HL1, *L. kefiranofaciens* HL1; ST 12268, *S. thermophilus* BCRC 12268; ST 13869, *S. thermophilus* BCRC 13869^T^; APL15, *Lc. lactis* subsp. *cremoris* APL15.

**Figure 4 foods-10-02098-f004:**
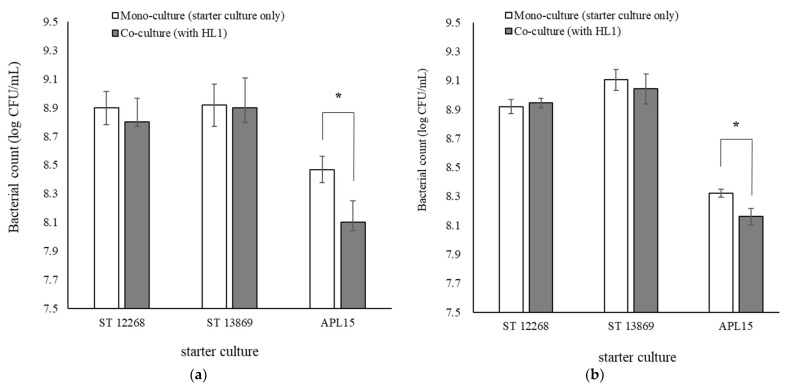
Effect of HL1 on bacterial counts of each selected co-culturing strain in skimmed milk at 30 °C for (**a**) 12 and (**b**) 24 h. The data are given as mean ± SD (*n* = 3). * Bars with a start within the same starter culture indicate a significant difference (*p* < 0.05). Abbreviations: HL1, *L. kefiranofaciens* HL1; ST 12268, *S. thermophilus* BCRC 12268; ST 13869, *S. thermophilus* BCRC 13869T; APL15, *Lc. lactis* subsp. *cremoris* APL15.

**Figure 5 foods-10-02098-f005:**
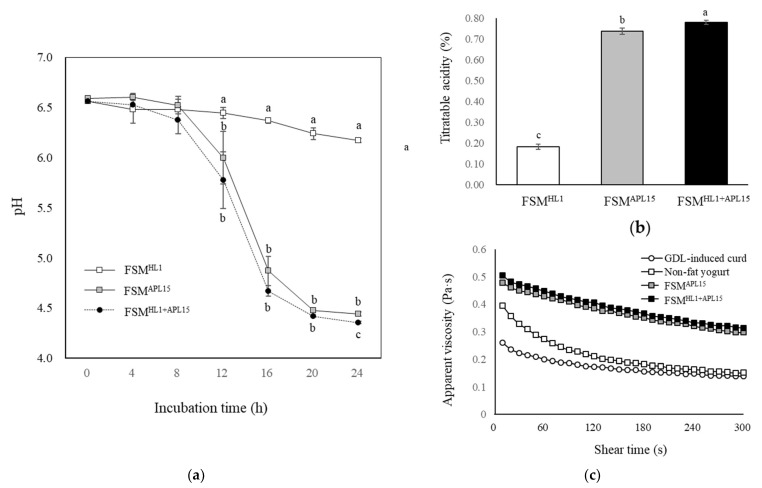
(**a**) The pH profile of fermented skimmed milk inoculated with different cultures during fermentation at 30 °C for 24 h. (**b**)Titratable acidity of fermented skimmed milk prepared by three kinds of cultures. The data are given as mean ± SD (*n* = 3). ^a–c^ Points without the common letter within the same incubation time and ^a–c^ bars without the common letter are significantly different (*p* < 0.05). (**c**) Apparent viscosity of four kinds of acid-induced milk curds (pH 4.50 ± 0.05) with shear time at constant temperature (4 °C) and rotation speed (30 rpm). FSM ^HL1^, fermented skimmed milk prepared by culturing with *L. kefiranofaciens* HL1 alone; FSM^APL15^, fermented skimmed milk prepared by culturing with *Lc. lactis* subsp. *cremoris* APL15 alone; FSM^HL1+APL15^, fermented skimmed milk prepared by co-culturing *L. kefiranofaciens* HL1 with *Lc. lactis* subsp. *cremoris* APL15; GDL-induced curd, the sample prepared by skimmed milk with 1.5% GDL; non-fat yogurt: sample prepared by culturing with commercial starter cultures (*S. thermophilus* and *L. delbrueckii* subsp. *bulgaricus*).

**Table 1 foods-10-02098-t001:** Syneresis and texture attributes of the fermented skimmed milk samples at pH 4.50.

Samples ^1^	Syneresis (%)	Firmness (g)	Consistency (g×s)	Cohesiveness (g)	Viscosity Index (g×s)
GDL-induced curd	44.89 ± 10.96 ^a^	15.23 ± 0.73	389.84 ± 3.47	8.66 ± 0.22	45.35 ± 3.30
Non-fat yoghurt	36.59 ± 2.22 ^a,b^	15.98 ± 0.59	395.66 ± 7.17	8.91 ± 0.13	50.36 ± 2.19
FSM^APL15^	27.59 ± 0.88 ^b^	15.63 ± 0.65	391.05 ± 5.06	9.52 ± 0.38	59.91 ± 8.48
FSM^HL1+APL15^	23.40 ± 2.55 ^b^	17.13 ± 1.13	402.92 ± 14.88	9.83 ± 0.83	54.22 ± 5.54

^1^ GDL-induced curd: sample prepared by skimmed milk with 1.5% GDL at 30 °C to pH 4.5; non-fat yogurt: sample prepared by culturing with commercial starter cultures (*S. thermophilus* and *L. delbrueckii* subsp. *bulgaricus*) at 30 °C to pH 4.5; FSM^APL15^: sample prepared by culturing with *Lc. lactis* subsp. *cremoris* APL15 alone at 30 °C to pH 4.5; FSM^HL1+APL15^: sample prepared by co-culturing *L. kefiranofaciens* HL1 with *Lc. lactis* subsp. *cremoris* APL15 at 30 °C to pH 4.5. Data are given as mean ± SD (*n* = 3). Means in the same column with different small letters are significantly different (*p* < 0.05).

**Table 2 foods-10-02098-t002:** Effect of starter cultures on sensory profiles of the fermented skimmed milk samples.

Fermented Skim Milk ^1^	Appearance	Aroma	Texture	Flavor	Overall Acceptability
Non-fat yogurt	6.29 ± 1.48	6.78 ± 1.45 ^a^	5.28 ± 1.96	5.78 ± 1.80 ^a^	5.88 ± 1.67 ^a^
FSM^APL15^	6.82 ± 1.10	6.05 ± 1.29 ^b^	5.78 ± 1.55	5.68 ± 1.90 ^a^	5.46 ± 1.78 ^a^
FSM^HL1+APL15^	6.72 ± 1.35	4.66 ± 1.75 ^c^	5.86 ± 1.45	4.09 ± 2.21 ^b^	4.60 ± 1.97 ^b^

^1^ Non-fat yogurt: sample prepared by culturing with commercial starter cultures (*S. thermophilus* and *L. delbrueckii* subsp. *bulgaricus*) at 30 °C to pH 4.5; FSM^APL15^: sample prepared by culturing with *Lc. lactis* subsp. *cremoris* APL15 alone; FSM^HL1+APL15^: sample prepared by co-culturing *L. kefiranofaciens* HL1 with *Lc. lactis* subsp. *cremoris* APL15 at 30 °C to pH 4.5. Data are given as mean ± SD (*n* = 30). Means in the same column with different superscript letters are significantly different (*p* < 0.05).

## Data Availability

The data presented in this article are available on reasonable request, from the corresponding author.

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
