# Peer review of "Co-Culture Strategy of Lactobacillus kefiranofaciens HL1 for Developing Functional Fermented Milk"

_foods, 2021, doi:10.3390/foods10092098_

Round 1
Reviewer 1 Report
The manuscript presents a co-culture strategy for cultivating probiotic strain L. kefiranofaciens HL1 and L. lactis APL15 (or other S. thermophilus) in milk to potentially make more functionalized dairy products. It is a good approach to stimulate the growth (enhance food functionalities) of a special probiotic strain, which can not grow well itself in dairy medium, with the help of starter cultures. Overall the results are clearly presented. But it requires some improvements.
- There is a problem for the differentiation method between HL1 and APL15. APL15 should also be able to grow in the acidified MRS medium, isn't it?
- The growth stimulation of HL1 by APL15 in skm is very small (almost nothing). It would be interesting in the following studies to find a real good partner for HL1.
- Is the poor growth of HL1 due to weak lactose metabolism? Is its genome sequence of HL1 accessible? Are there lactose operon or lactose metabolism genes located in the genome or plasmid? It will benefit the readers to discuss these aspects regarding the poor growth of HL1.
Reviewer 2 Report
Interesting paper.
However the authors should carry out a more detailed literature review and mention recent papers on fermented foods.
Other than that results are well described and discussed.
Reviewer 3 Report
This is an interesting work indeed. There is an increasing demand for probiotic products, and we need more novel probiotic products in the market. Hence this work is timely.
Introduction
Lines 48-66 : In here these authors have explained only the beneficial influence of probiotic co-culture. That is true however there are some reports on non-significant influence of probiotic co-culturing on sensory quality improvements in dairy foods. I think authors must appreciate and include such information very briefly in here as well. In that way the introduction will provide a balanced account on what is available on the literature. Does not matter whether is bacteria-yeast or bacteria-bacteria co-culturing but the appreciation of such effects is important and highly recommended in here. This will definitely help to further improve this manuscript.
An example paper to read and cite as appropriate.
Ranadheera, C. S., Evans, C. A., Adams, M., & Baines, S. K. (2016). Co-culturing of probiotics influences the microbial and physico-chemical properties but not sensory quality of fermented dairy drink made from goats’ milk. Small Ruminant Research, 136, 104-108.
Lines 68-79: Could you please include one or two brief sentences t explain the novelty of this product or how it is different t products available in the market or in previous research in the literature?
In the materials and methods: 2.4.2. Sensory Evaluation
More information is needed here. What is the size of your sensory panel (how many members)? Trained untrained or semi-trained ? Their age range ? How many males and females in the panel? How did you present your samples / Where did you conduct this sensory evaluation? Is this evaluation carried out on fresh product or stored product ? Have you received ethical approval from a relevant body / university for this work ? Is so what is the ethics approval number ?
Figures and tables look very clear and appropriate. It explain the findings very well. Discussion is based on the findings / results and it connects well. Conclusion seems appropriate.
Round 2
Reviewer 3 Report
It is apparent that the authors have addressed my comments carefully and thoroughly. It improved the manuscript significantly and can be highly recommended for publication. I thank authors for their careful attention and editors for giving me this opportunity t review this work.